# A Compact Microwave-Driven UV Lamp for Dental Light Curing

**Siyuan Liu, Yuqing Huang and Qinggong Guo ***

College of Electronic and Information Engineering, Sichuan University, Chengdu 610065, China; 2020141450186@stu.scu.edu.cn (S.L.); huangyuqing@stu.scu.edu.cn (Y.H.)
* Correspondence: guoqingong@scu.edu.cn; Tel.: +86-28-8547-0659

**Abstract:** The size of current microwave-driven UV lamps limits their direct application in dental light curing. This article proposes a coaxial structure to miniaturize the UV lamp. First, the Drude model and the finite difference time domain algorithm were used to analyze the multi-physical field coupling and the complex field distribution within the lamp. Second, the dimensional parameters of the lamp were optimized, which enabled the lamp to be miniaturized and operate with high performance. Third, to analyze the sensitivity of the lamp, the effects of input power, gas pressure, and gas composition on its performance were investigated. It was found that an input power of 6 watts was enough to light the bulb with over 90% energy utilization. Finally, to verify the feasibility, an experimental system was set up. The lamp was successfully lit in the experiment, and its spectral output was tested. The results show that the microwave-driven UV lamp based on a coaxial structure is miniaturized and broad-spectrum, making it suitable for clinical dental light curing.

**Keywords:** dental light curing; microwave-driven UV lamp; miniaturization; Drude mode

## 1. Introduction

Dental light curing is a crucial step in dental restorative procedures, such as filling tooth cavity liners, pits, cracks, and root canal posts, to ensure the longevity and durability of the restoration. A light curing lamp should have a broad spectrum and a high energy output [1]. The absorption spectrum of common dental resin materials ranges from 360 nm to 485 nm, so the curing lamp should emit a wide spectrum to adapt to different resin materials [2–6]. Additionally, insufficient light output can lead to inadequate curing, which may cause secondary restorative failure [4,7]. Therefore, curing lamps need to have a high luminous efficiency. Ultraviolet (UV) light is a suitable method for dental curing, as it meets both of these requirements.

There are two common types of lamp for dental UV curing: UV LEDs and plasma arc lamps [8–11]. LEDs are small and portable, but they have a narrow range of light waves and can only cure a few kinds of resin materials [12]. Plasma arc lamps have a broader spectrum and higher power density, achieving a better curing effect under equal conditions. However, they have a short lifespan about 40 to 100 h and require frequent maintenance [13].

A microwave-driven electrodeless UV lamp is an ideal and efficient light source for dental curing, as it combines the strengths of traditional curing lamps while avoiding their limitations [14]. It has a high power density, broad spectrum, long lifespan, and stable output. However, early microwave-driven electrodeless plasma lamp devices were large and unreliable due to their working mechanisms, which involved magnetrons, resonant cavities, quartz bulbs, etc. To miniaturize the lamps, some studies have been carried out. A common method has been to replace metal waveguides with dielectric resonant cavities [15–17]. Hafidi et al. reduced the resonant cavity size by loading a thin metal plane in the cavity to increase the equivalent capacitive reactance and reduce the resonant frequency [18]. Espiau et al. designed a compact coaxial resonant cavity for driving the

lamp [19], and Jiade Yuan et al. further scaled down the size by loading metal plates in the coaxial cavity [20]. The plasma lamp designed by Topanga featured a compact air resonator and an original ground-coupled structure with an integrated lamp assembly, minimizing size while reducing cost and improving performance [21]. Moreover, C. Ferrari et al. used coaxial antenna excitation instead of MW cavity to excite plasma discharge in the bulb, resulting in a more compact structure and higher efficiency [22]. However, these methods were not suitable for the design of UV lamps used in dental light curing, which required further miniaturization of the equipment.

This study proposes a coaxial structure for the microwave-driven UV lamp for dental light curing. The lamp is designed in a concave cylindrical shape and is inserted into the end of the coaxial structure, so that it is located at the center of the maximum electric field. This design enables the miniaturization of the microwave-driven UV lamp while ensuring the excitation effect. In the next section, multi-physical field coupling is analyzed based on the relevant principles of electromagnetic theory. The Drude model is used to analyze the plasma formed by ionization of the argon–mercury (Ar-Hg) gas mixture. The finite difference time domain (FDTD) algorithm is also applied in the analysis. Then, the dimensions of the lamp are optimized and adjusted, and the performance of the model is verified experimentally. Furthermore, the effects of some parameters, such as input power, gas pressure, and gas components, are investigated. Finally, an experimental system is built, and the lamp is successfully lit. The results show that the microwave-driven UV lamp achieves compactness of construction and has a wide spectrum. It is proven that this compact plasma lamp can be used for clinical dental light curing.

## 2. Materials and Methods

### 2.1. Geometry

The coaxial structure has the advantage of compactness and transmission of high-frequency electromagnetic waves, so this article uses this structure for the design of the microwave-driven UV lamp. Figure 1 shows the structural schematic of the lamp.

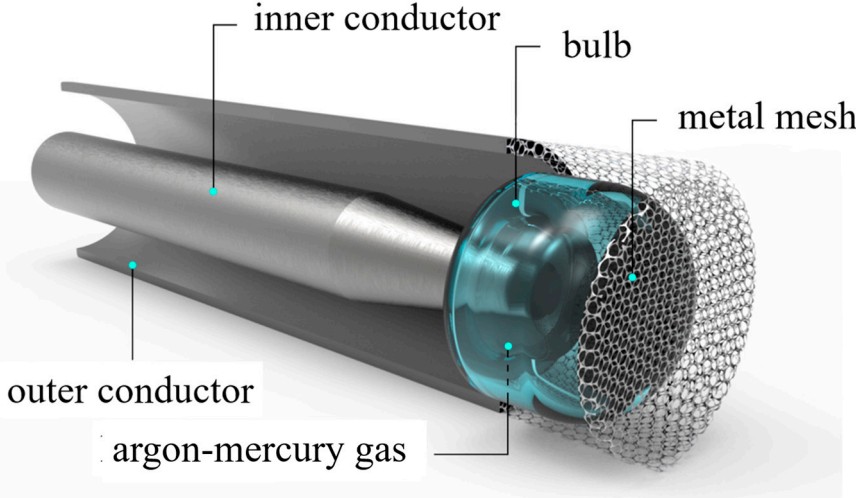

**Figure 1.** Structure diagram of miniaturized induction lamp.

The bulb is filled with an Ar-Hg gas mixture and inserted in a nested pattern between the inner and outer conductors of the coaxial end. The end of the inner conductor is tapered to ensure the proper insertion of this small bulb for practical processing reasons. Figure 2 displays the dimensional view of the lamp.

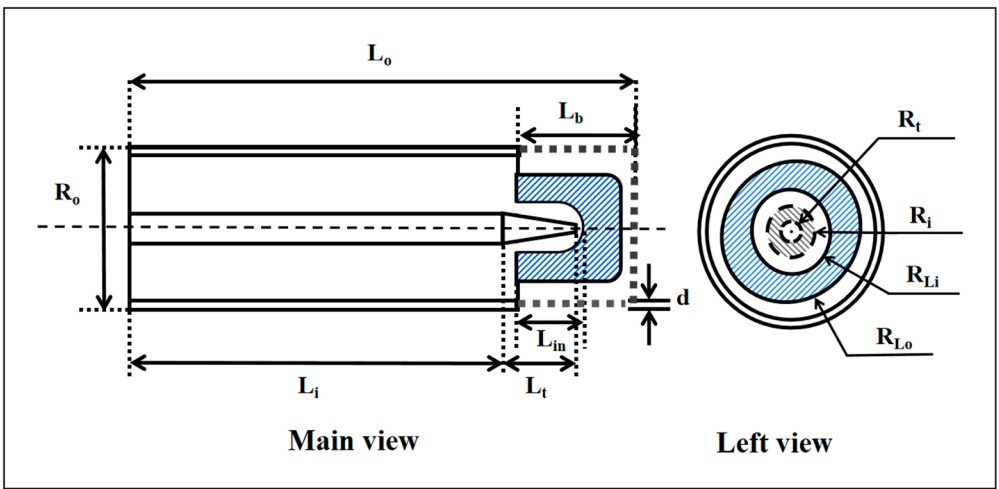

**Figure 2.** A 2D dimensional view of the lamp.

To further enhance the prevention of electromagnetic wave leakage, a metal mesh is incorporated at the end of the lamp, as illustrated in Figure 1. The metal mesh has a maximum diameter of less than 8 mm, effectively shielding out 2.45 GHz electromagnetic waves. Importantly, the metal mesh maintains a strong light transmittance, ensuring minimal impact on the lamp's output.

In order to validate the shielding effectiveness of the metal mesh, separate tests were conducted to examine the transmission efficiency when adding or omitting the metal mesh within a section of the coaxial structure. The results are depicted in Figure 3, where $S_{11}$ and $S_{21}$ are scattering parameters to quantify the reflection and transmission, respectively, of the incident power in the UV lamp system. These parameters play a crucial role in determining the transmission efficiency of the system.

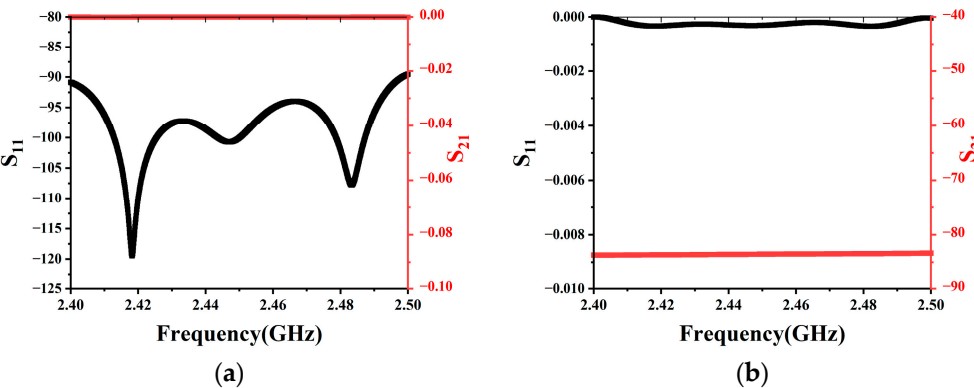

**Figure 3.** Comparison of $S_{11}$ and $S_{21}$: (**a**) without metal mesh (**b**) with metal mesh.

When losses are not significant or negligible, based on these two parameters, the transmission efficiency of the system $\eta$ can be calculated by the equations below, where $S_{11}$ and $S_{21}$ are the values in decibels and $\Gamma$ represents the reflection coefficient.

$$\Gamma = 10^{\frac{S_{11}}{20}}, \tag{1}$$

$$\eta = \frac{|S_{21}|^2}{1 - |\Gamma|^2}, \tag{2}$$

Based on the results in Figure 3, it is observed that, without the metal mesh, the $S_{11}$ is less than $-90$ dB and the $S_{21}$ value is about 0 dB, which means the transmission efficiency of

the coaxial structure without a metal mesh is more than 99.98%. In contrast, with the metal mesh added, the $S_{11}$ is approximately $-0.0005$ dB and the $S_{21}$ is around $-84.3$ dB, which means the transmission efficiency is about 0.0196%. It can be concluded that incorporating a metal mesh effectively isolates electromagnetic wave leakage.

*2.2. Microwave Theory*

As high-frequency electromagnetic waves, microwaves are used to excite and sustain plasma in plasma lamps. Combining Maxwell's equations and the theory of plasma electron drift, the interaction between microwaves and plasma can be simply described as the coupling of plasma parameters and conductivity.

The electromagnetic field is described by Maxwell's Equation (3), which can be expressed as follows:

$$
\begin{aligned}
\nabla \times \vec{E} &= -\frac{\partial}{\partial t}\vec{B} \\
\nabla \times \vec{H} &= \vec{J_P} + \frac{\partial}{\partial t}\vec{D} \\
\nabla \cdot \vec{B} &= 0 \\
\nabla \cdot \vec{D} &= \rho
\end{aligned}
\tag{3}
$$

where $\vec{E}$ is the electric field vector, $\vec{B}$ is the flux density vector, $\vec{H}$ is the magnetic field vector, $\vec{D}$ is the electric displacement intensity vector, $\vec{J_P}$ is the plasma current density vector, and $\rho$ is the free charge density.

According to Ohm's law, there is Equation (4):

$$
\vec{J_P} = \sigma \vec{E}
\tag{4}
$$

where $\sigma$ is the conductivity, a measure of how easily charges move through it, and resistivity $\rho$ is the reciprocal of conductivity $\sigma$ shown in Equation (5).

$$
\rho = \frac{1}{\sigma}
\tag{5}
$$

The classical equation for plasma conductivity is given as Equation (6).

$$
\sigma = \frac{n_e e^2}{m_e v_m}
\tag{6}
$$

where $m_e$ is the electron mass, $e$ is the meta-charge, $n_e$ is the electron density, and $v_m$ is the effective constant collision frequency, a kind of plasma parameter.

*2.3. Plasma Parameter Settings*

Plasma is a state of matter mainly composed of free electrons and charged ions. When the rare gases filled in the bulb are excited by 2.45 GHz high-frequency electromagnetic waves, they collide with each other at a high frequency, resulting in the ionization of a large number of free electrons and charged ions and the formation of a plasma. Under the frequency of 2.45 GHz, the variation in electron mobility is extremely small and negligible, i.e., the plasma parameters can be considered as constants. To determine the plasma parameters, the Drude model is used for analytical calculations, which includes three key parameters: the breakdown field strength of rare gases, plasma frequency, and collision frequency.

The breakdown field strength of the rare gas is calculated using the following Equations (7) and (8) [23,24].

$$
Ec = \frac{K_B T \omega}{pS\Lambda\sqrt{\frac{m_e v_i}{3e}}},
\tag{7}
$$

$$\Lambda = \frac{1}{\sqrt{\left(\frac{\pi}{l}\right)^2 + \left(\frac{2.405}{R}\right)^2}}, \tag{8}$$

In Equation (1), $K_B$ is Boltzmann's constant, which is equal to $1.38 \times 10^{-23}$ J/K, $T$ denotes the temperature of the rare gas, $\omega$ is the excitation frequency, $p$ is the gas pressure, $v_i = 15.8$ eV represents the first ionization energy of the neutral particle of the filled gas, and $S$ denotes the cross-sectional area of the neutral particle at elastic collision. $\Lambda$ is the characteristic diffusion length in Equation (2). $l$ and $R$ are the axial length and radius of the plasma in the bulb, respectively. Then, the critical plasma electric field is approximated as $3.98 \times 10^3$ V/m. That is, the moment the electric field strength within the bulb exceeds the critical value, rare gases undergo collisions, become ionized, and emit ultraviolet light.

Afterwards, the plasma frequency $\omega_{pe}$ can be calculated from the following Equation (9).

$$\omega_{pe} = \sqrt{\frac{n_e e^2}{\varepsilon_0 m_e}}, \tag{9}$$

where $\varepsilon_0$ is the vacuum permitivity.

After that, the collision frequency, $v_m$, is calculated using Equation (10).

$$v_m = \frac{\sqrt{8 K_B T_e / \pi m_e}}{\lambda_e}, \tag{10}$$

where $T_e$ denotes the electron temperature and $\lambda_e$ is the average free distance between the neutral atom and the electron.

Once the plasma frequency $\omega_{pe}$ and collision frequency $v_m$ are known, the relative permittivity $\varepsilon_p$ can be calculated using Equation (11), with $E'_{ps}$ and $E''_{ps}$ being its real and imaginary parts, respectively.

$$\varepsilon_p = \varepsilon_0 \left( 1 - \frac{\omega^2_{pe}}{\omega(\omega - j v_m)} \right) = E'_{ps} - E''_{ps}, \tag{11}$$

*2.4. Input Parameters and Boundary Conditions*

In most applications, both inner and outer conductors typically have high electrical conductivity and provide good shielding against electromagnetic waves. The simulations assume that the default inner and outer conductors are perfect electric conductor materials. The boundary conditions are given by Equations (12) and (13), which indicate that, inside the coaxial structure, the tangential component of the electric field and the normal component of the magnetic field are zero. It can be approximated that the electromagnetic wave is fully emitted and forms a standing wave inside.

$$\vec{e_t} \times \vec{E} = 0, \tag{12}$$

$$\vec{e_n} \times \vec{H} = 0, \tag{13}$$

where $\vec{e_t}$ is the tangential component of the electric field and $\vec{e_n}$ is the normal component of the magnetic field.

In addition, the vital input parameters in the simulation are shown in Table 1.

**Table 1.** Input plasma parameters.

| Parameter | Value |
|---|---|
| Input power | 6 W |
| Microwave frequency | $2.45 \times 10^9$ Hz |
| Plasma frequency | $3.4 \times 10^{11}$ rad/s |
| Collision frequency | $3 \times 10^8$ Hz |
| Field breakdown | $3.98 \times 10^3$ V/m |
| Plasma frequency maintained | $1.7 \times 10^{11}$ rad/s |

## 3. Results and Discussion

### 3.1. Dimension Optimization

This section optimizes the dimensional parameters of the lamp and calculates the electric field distribution and the $S_{11}$ in this structure of the lamp using the FDTD method and the Drude model.

The FDTD method is a widely adopted numerical computational technique that offers improved accuracy in characterizing the propagation and coupling characteristics of electromagnetic fields. It is particularly useful for studying non-metallic plasma systems and analyzing nonlinear effects, including the dielectric response and nonlinear behavior of plasma when subjected to high-power excitation. This method plays a vital role in investigating plasma lamps operating under high-frequency and high-power conditions, enabling a deeper understanding of their performance. The principle is based on Maxwell's system of equations and the discretization of the electromagnetic field in space.

In the FDTD method, space is divided into grid cells and time is discretized into successive time steps. Through iterative computation, the time-domain behavior of the electromagnetic field can be simulated and information such as the propagation path, amplitude, and phase of the electromagnetic wave can be obtained. In this study, small hexahedral grid cells are used with an accuracy of $-40$ dB. The step size is determined by the minimum grid size, which is set to 0.101021 mm in this case.

Furthermore, the Drude model is a classical physical model that simplifies the conductive behavior of a plasma, making it an efficient choice for designing electrodeless plasma lamps. Despite its simplifications, it has been widely used in plasma physics research and has shown relatively accurate results in specific scenarios [25–27]. Compared to other models like particle dynamics models and fluid models, it strikes a balance between simplicity and accuracy. It allows for initial predictions regarding the efficiency and power consumption of UV lamps, providing a solid starting point for further design optimization. At higher frequencies like those of the optical band, additional effects such as dielectric polarization and plasmon resonance may occur in the plasma. These effects require more complex theoretical models for accurate description. However, since this study focuses on a frequency range much smaller than terahertz frequencies, utilizing the simpler Drude model is sufficient for analysis.

The optimization criteria include three main aspects: (a) the electric field strength should be higher than the gas breakdown electric field strength to ensure that the rare gas inside the bulb can be ionized to produce UV light, and (b) the transmission efficiency of the system should be above 90%, which means that $S_{11}$ should be less than $-10$ dB, as indicated by Equations (1) and (2), and (c) the electric field strength distribution inside the bulb should be uniform, which means that the value of the electric field uniformity COV should be no bigger than 1.0. COV is expressed in Equation (14),

$$COV = \sqrt{\frac{\sum_n \left(E_i - \overline{E}\right)^2}{n}} / \overline{E}, \tag{14}$$

where $E_i$ is the value of the electric field at the ith point in the selected region, $\overline{E}$ is the average value of the electric field in the selected region, and $n$ is the total number of points in the selected region.

The first parameters to determine were the inner and outer radius of the coaxial structure. Since the common laboratory impedances are 50 ohms, the inner conductor radius of the coaxial line was set to 3 mm and the outer conductor radius to 7 mm.

Then, the effects of the dimensional parameters such as the length of the inner conductor ($L_i$), the length of the frustum ($L_t$), the insertion depth of the bulb ($L_{in}$), and its inner and outer radius ($R_{Li}$ and $R_{Lo}$) were analyzed and the preferred parameter values were obtained. Figure 4 shows the effects of these parameters on $S_{11}$.

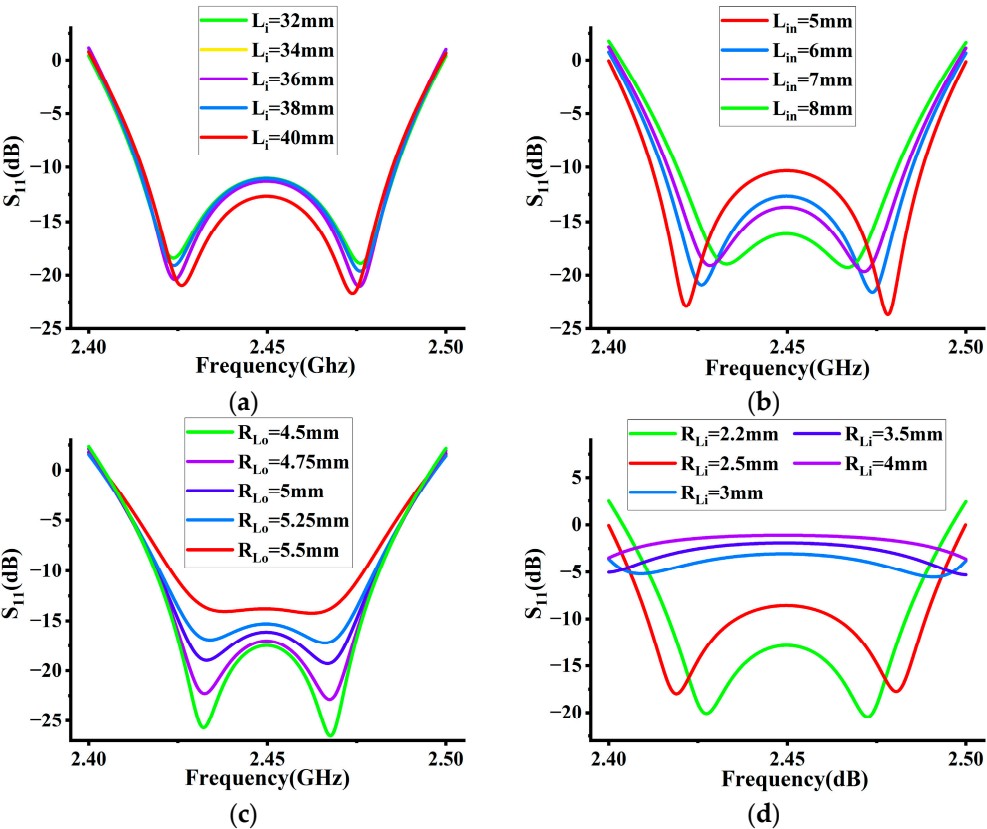

**Figure 4.** (**a**) The effect of $L_i$ on $S_{11}$; (**b**) the effect of $L_{in}$ on $S_{11}$; (**c**) the effect of $R_{Lo}$ on $S_{11}$; (**d**) the effect of $R_{Li}$ on $S_{11}$.

At 2.45 GHz, a smaller $S_{11}$ means higher energy utilization. Therefore, from the analysis of the above-mentioned parameters, the final microwave-driven UV lamp dimensional parameters optimized for this experiment were obtained, as shown in Table 2 below. Figures 5 and 6 show the final calculated $S_{11}$ and electric field distribution for the optimized model.

It can be observed from Figure 5 that $S_{11}$ is less than $-13$ dB, indicating a near 100% energy utilization efficiency. Additionally, it can be seen from Figure 6 that the majority of the bulb region is subjected to a uniform electric field, exceeding the breakdown electric field of the filling gas. This means that it is possible to ionize the inert gas and emit ultraviolet rays.

**Table 2.** Input dimensional parameters.

| Parameter | Definitions | Value |
|---|---|---|
| $R_i$ | Radius of inner conductor (Radius of the bottom plane of the frustum) | 3 mm |
| $R_o$ | Radius of outer conductor | 7 mm |
| $R_t$ | Radius of the upper plane of the frustum | 0.8 mm |
| $R_{Li}$ | Inner radius of the bulb | 2.2 mm |
| $R_{Lo}$ | Outer radius of the bulb | 4.5 mm |
| d | Thickness of outer conductor | 0.3 mm |
| $L_i$ | Length of inner conductor | 40 mm |
| $L_o$ | Length of outer conductor | 53.5 mm |
| $L_t$ | Length of the frustum | 12 mm |
| $L_{in}$ | Depth of bulb insertion | 8 mm |
| $L_b$ | Length of the overall bulb | 9 mm |

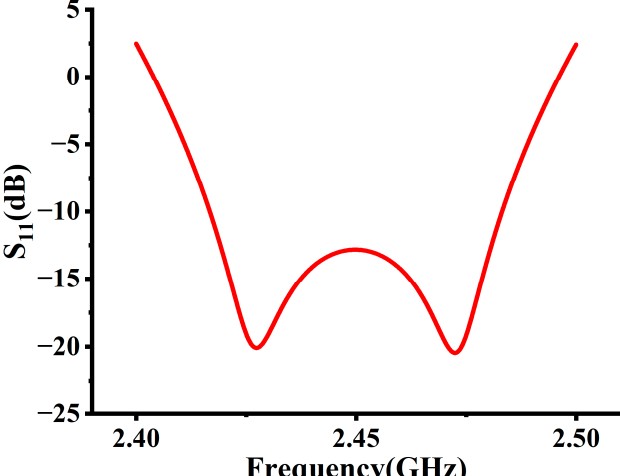

**Figure 5.** $S_{11}$ of optimized model.

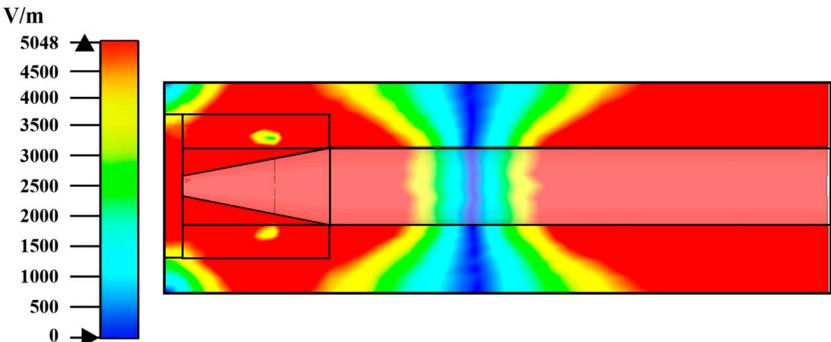

**Figure 6.** Electric field distribution of optimized structure.

### 3.2. Plasma Parameter Sensitive Analysis

3.2.1. Input Power Analysis

In this section, the effects of different input powers on the electric field and energy utilization in the lamp are discussed. Using the FDTD algorithm, the input power was varied to 4, 6, 8, and 12 watts, and the $S_{11}$ curves shown in Figure 7 and the comparative electric field distribution in Figure 8 were obtained. The average values of electric field strength in bulb region and COV at different input power levels are shown in Table 3.

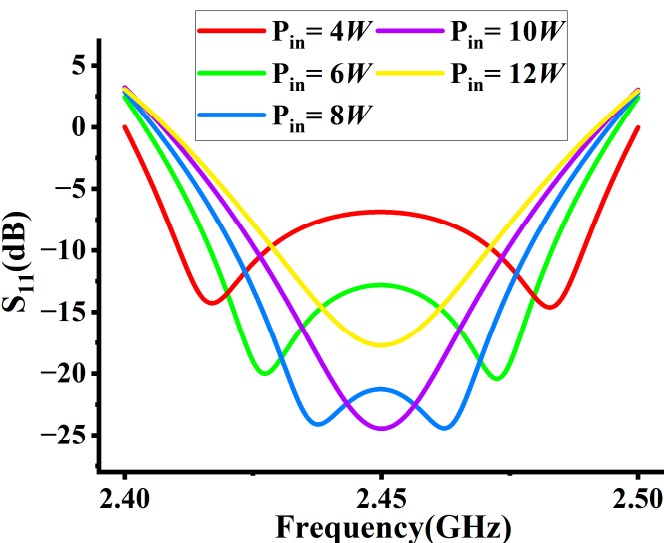

**Figure 7.** The effect of input power $P_{in}$ on $S_{11}$.

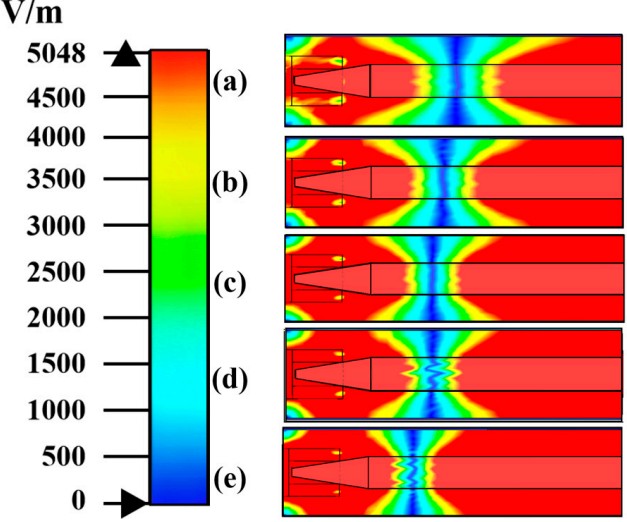

**Figure 8.** Electric field distributions at different input power $P_{in}$: (**a**) $P_{in}$ =4 W, (**b**) $P_{in}$ =6 W, (**c**) $P_{in}$ = 8 W, (**d**) $P_{in}$ = 10 W, (**e**) $P_{in}$ = 12 W.

**Table 3.** Average values of electric field strength in bulb region and COV at different input power.

| Input Power | COV | Average Values of Electric Field Strength in Bulb Region |
|---|---|---|
| 4 W | 0.97 | 8897 V/m |
| 6 W | 1.00 | 11,218 V/m |
| 8 W | 1.05 | 13,095 V/m |
| 10 W | 1.07 | 14,615 V/m |
| 12 W | 1.09 | 15,874 V/m |

From the results, the $S_{11}$ is less than $-10$ dB except when $P_{in}$ = 4 W, and the energy utilization is at least 90%. That is, the minimum input power of the lamp is 6 W.

As the input power increases, $S_{11}$ at 2.45 GHz shows a trend of becoming smaller in the range of 4~10 W, and then $S_{11}$ at 12 W becomes larger. From the theoretical analysis, in the range of 4~10 W, as the input power increases, the electrons gain more energy from the electric field, and the chance of collision with other particles increases greatly, resulting in more excitation ionization, so the $S_{11}$ will become smaller. When the power continues to

increase, the excess input energy is wasted instead, so the energy utilization rate decreases and the $S_{11}$ increases.

Combining Figure 8 and Table 3, it can be concluded that the electric field strength in the bulb region exceeds the breakdown strength of the gas in all five cases. Furthermore, as the power increases from 4 W to 12 W, the corresponding electric field strength increases accordingly, but the electric field uniformity decreases accordingly. Only when $P_{in}$ = 4 W and 6 W is the electric field distribution considered uniform.

### 3.2.2. Gas Pressure Analysis

In the case of a bulb filled with the same gas, the gas pressure is an important factor affecting the collision frequency of the plasma $v_m$ and the gas breakdown voltage. As the gas pressure increases, the mean free path of electrons decreases and the collision frequency $v_m$ increases linearly, resulting in different values of the gas breakdown electric field in the same circumstances [28]. This section discusses the electric field distribution and energy utilization inside the lamp at different gas pressures where $p$ = 20 pa, 50 pa, 100 pa, 150 pa, and 200 pa. The results are shown in Figures 9 and 10 and Table 4.

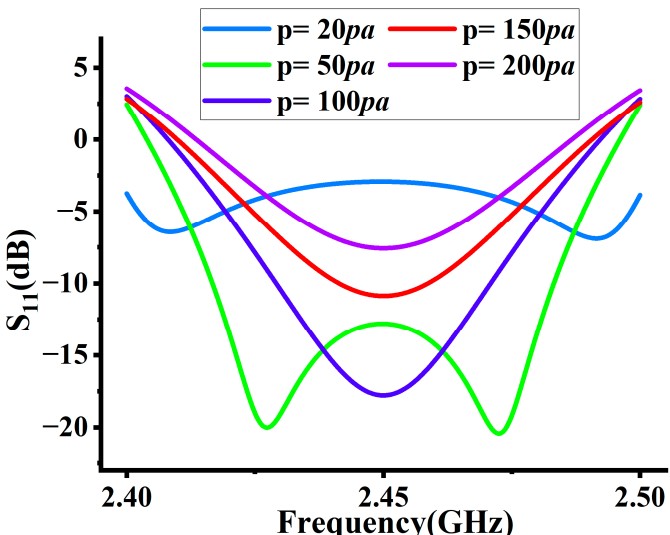

**Figure 9.** The effect of gas pressure $p$ on $S_{11}$.

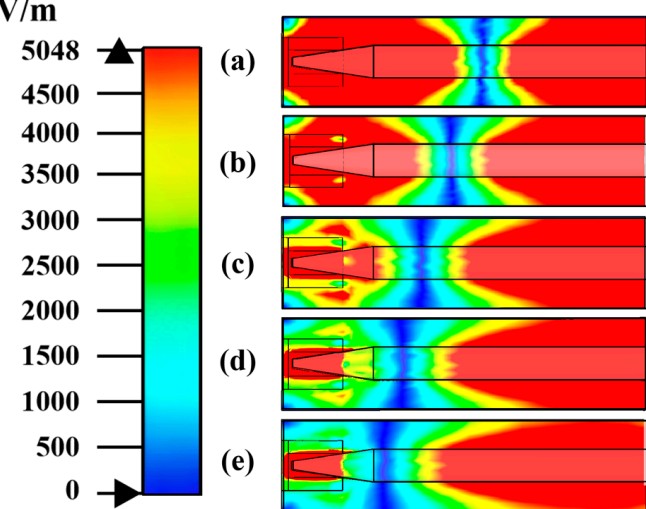

**Figure 10.** Electric field distributions at different gas pressure $p$: (**a**) $p$ =20 pa, (**b**) $p$ = 50 pa, (**c**) $p$ = 100 pa, (**d**) $p$ = 150 pa, (**e**) $p$ = 200 pa.

**Table 4.** Average values of electric field strength in bulb region and COV at different gas pressures.

| Gas Pressure | COV | Average Values of Electric Field Strength in Bulb Region |
|---|---|---|
| 200 pa | 1.09 | 5214 V/m |
| 150 pa | 1.09 | 6050 V/m |
| 100 pa | 1.10 | 7954 V/m |
| 50 pa | 1.00 | 11,218 V/m |
| 20 pa | 0.86 | 16,827 V/m |

In the results in Figure 9, the $S_{11}$ is less than $-10$ dB at $p = 50$ pa, 100 pa, 150 pa, and the energy utilization reaches 90%. With the increase in gas pressure, the $S_{11}$ at 2.45 GHz shows a tendency to become smaller in the range of 20~100 pa, and then larger at 100~200 pa.

Theoretically, the gas pressure has a non-linear effect on the UV radiation of the lamp. When the gas pressure is low, as the gas pressure increases, the average free range of electrons decreases, the electron collision frequency $v_m$ increases, and more atoms are excited to ionize after collision with electrons. However, as the pressure continues to rise, the possibility of electron–atom collisions increases, and the energy lost by electrons increases correspondingly. Combined with the fact that the average free range of electrons is too short, this will lead to a decrease in the energy gained by the electrons from the electric field, causing the final energy utilization rate to decrease and $S_{11}$ to become larger.

Combining the data from Figure 10 and Table 4, the electric field distribution is uniform only when $p = 20$ pa and 50 pa. In all five cases, the average values of the electric field strength in the bulb region are greater than the gas breakdown strength, implying that the gas in the bulb can be ionized and radiate UV light.

### 3.2.3. Gas Component Analysis

The parameters of the generated plasma are different depending on the component of gas filled in the bulb. Various gas components exhibit distinct ionization energies, resulting in different effective collision frequencies and values of the gas breakdown electric field in identical circumstances [29]. In the same case, six different gases were selected for analysis with the Drude model, and the results of the $S_{11}$ and electric field distribution for different gas types were obtained as shown below in Figures 11 and 12 and Table 5.

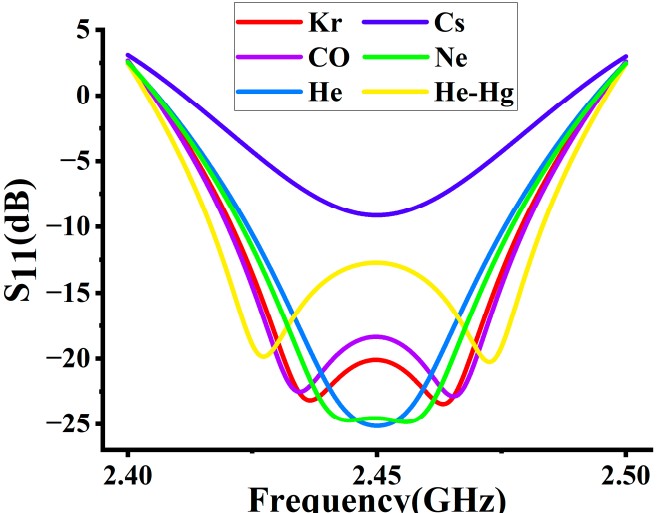

**Figure 11.** The effect of gas component on $S_{11}$.

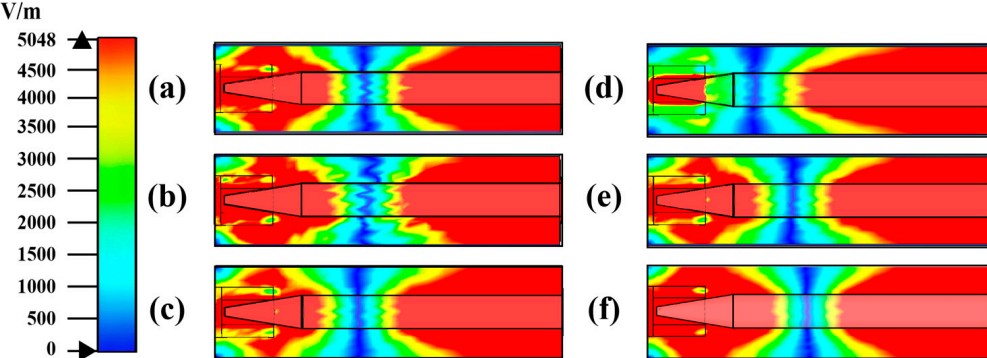

**Figure 12.** Electric field distributions at different gas component: (**a**) Kr, (**b**) CO, (**c**) He, (**d**) Cs, (**e**) Ne, (**f**) Ar-Hg.

**Table 5.** Average values of electric field strength in bulb region and COV at different gas complement.

| Gas Complement | COV | Average Values of Electric Field Strength in Bulb Region |
|---|---|---|
| Ar-Hg | 1.00 | 11,218 V/m |
| Kr | 1.02 | 11,165 V/m |
| CO | 1.01 | 11,171 V/m |
| He | 1.08 | 8896 V/m |
| Ne | 1.07 | 9259 V/m |
| Cs | 1.09 | 5730 V/m |

From Figure 11, it is found that under the same conditions, $S_{11}$ is larger than $-10$ dB when Cs gas is filled, and smaller than $-10$ dB when other gases are filled. The gas types with $S_{11}$ from large to small are Ar-Hg, CO, Kr, Ne, and He.

In terms of the electric field distribution, by combining Figure 12 and Table 5, it can be concluded that the average values of each field strength are larger than the breakdown strength values of the corresponding gases, but the electric field distribution is uniform only when filled with Ar-Hg gas.

The gas chosen in this experiment was the Ar-Hg gas mixture from the results; $S_{11}$ and the electric field strength were in line with the requirements.

### 3.3. Experimental Verification

A general schematic of the experimental system is shown in Figure 13. The magnetron source is used to provide microwaves. The circulator enables the electromagnetic waves to be transmitted in a unidirectional loop.

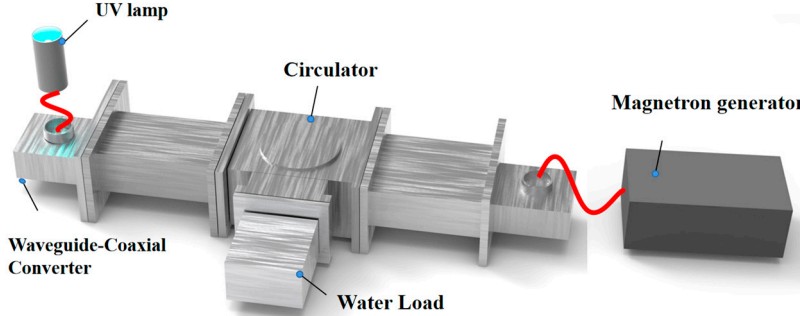

**Figure 13.** Diagram of the micro-driven UV lamp.

The lamp was successfully lit by high-frequency microwaves, as shown in Figure 14. Additionally, dental curing lamps typically feature a light outlet diameter ranging from 8 to 10 mm. This miniaturized lamp has an outlet diameter of 9 mm, which meets the size requirements for dental curing lamps.

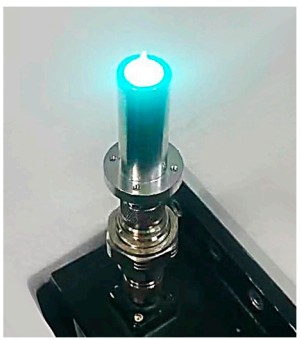

**Figure 14.** Actual light up.

In order to assess the spectral output of the lamps, the MAYA 2000PRO spectrograph was used for precise measurements. The UV light source was analyzed across a wavelength range of 250–600 nm.

Measurements were performed in a controlled environment consisting of a darkroom with a temperature of 25 degrees Celsius and a relative humidity of 50%. The relative spectral distribution of the measured UV lamps in the wavelength range is shown in Figure 15 below.

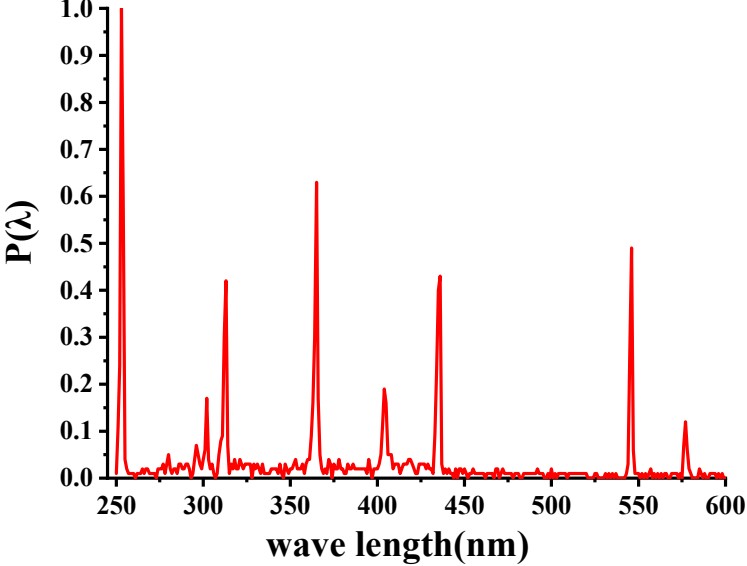

**Figure 15.** Relative spectral distribution diagram of the induction lamp.

The relative spectral distribution diagram shows the change in radiation intensity or power of the light source at different wavelengths, revealing the radiation characteristics and energy distribution of the light source. From Figure 15, it is found that the UV light radiated by the induction lamp reaches the maximum radiation intensity at 253 nm and has several peaks of radiation intensity in the wavelength range of 250~600 nm, indicating a broad spectrum.

## 4. Conclusions

This study presents a compact microwave-driven UV lamp for dental light curing, using the unique coaxial structure. Its exceptional contribution is to downsize the microwave electrodeless lamp for dental curing, which is a breakthrough in the dimensional limitation of traditional devices. The FDTD algorithm and the Drude model were used to optimize the structure. Then, the corresponding experimental system was built for verification. The feasibility of the compact structure of the lamp was demonstrated experimentally, and the small lamp was successfully lit. Its spectral output was also calculated, indicating a

broad spectrum. It is characterized by miniaturization, which means it can be used in dental curing surgery. Additionally, it provides a reference for the future optimization of microwave-driven UV lamp.

**Author Contributions:** Writing—original draft preparation, S.L.; writing—review and editing, Q.G.; visualization, Y.H.; project administration, Q.G.; funding acquisition, Q.G.; supervision, Q.G. All authors have read and agreed to the published version of the manuscript.

**Funding:** This research received no external funding.

**Data Availability Statement:** Not applicable.

**Conflicts of Interest:** The authors declare no conflict of interest.

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
