# Peer review of "A Compact Microwave-Driven UV Lamp for Dental Light Curing"

_processes, doi:10.3390/pr11092651_

Round 1
Reviewer 1 Report
Siyuan Liu et al. investigated a compact microwave-driven UV lamp for dental light-curing.
The current version already shows very high quality, both data and figure quality. Based on my current knowledge, I don’t have further comments. I recommend the acceptance of the current version.
Reviewer 2 Report
Comments attached.

Reviewer 3 Report
I believe that the miniaturization of the system proposed by the authors is a crucial research topic in the field of dental treatment. From that perspective, I would like to raise the following questions:
1. If the objective is to miniaturize the lamp, shouldn't one consider an operational structure? Given the inclusion of a microwave input device, doesn't this impose a limitation on the actual miniaturization?
2. Is there no possibility that the microwaves might adversely affect the actual cells?
3. While the data on the electric field seems comprehensive, isn't there a lack of actual UV control data?
4. Do you have data on the input energy versus UV output?
5. The title of Figure 14 needs to be revised.
Round 2
Reviewer 2 Report
File attached.

Reviewer 3 Report
The areas that seemed lacking in between appear to have been well complemented. I believe that content deemed necessary for the expansion of the technology's application has been appropriately added.